# Using a B-Phycoerythrin Extract as a Natural Colorant: Application in Milk-Based Products

**DOI:** 10.3390/molecules26020297

**Published:** 2021-01-08

**Authors:** Ana Belén García, Eleonora Longo, Mª Carmen Murillo, Ruperto Bermejo

**Affiliations:** 1Department of Physical and Analytical Chemistry, High Engineering Polytechnic School of Linares, University of Jaen, 23700 Linares, Spain; abgr0013@red.ujaen.es (A.B.G.); mcmc0031@red.ujaen.es (M.C.M.); 2Department of Pharmaceutical and Pharmacological Sciences, University of Padova, 35131 Padova, Italy; eleonora.longo@studenti.unipd.it

**Keywords:** B-phycoerythrin, colorimetry, milks, natural food colorants, color stability

## Abstract

Nowadays, there is a growing interest in finding new coloring molecules of natural origin that can increase and diversify the offer of natural food dyes already present in the market. In the present work, a B-phycoerythrin extract from the microalgae *Porphyridium cruentum* was tested as a food colorant in milk-based products. Using spectroscopy and colorimetry, the extract was characterized and gave evidence of good properties and good stability in the pH range between 4.0 and 9.0. Coloring studies were conducted to demonstrate that samples carrying the pink extract could be used for simulating the pink color of marketed milk-based products. The staining factors, representing the amount of pink protein to be added to reproduce the color of strawberry commercial products, ranged between 1.6 mg/L and 49.5 mg/L, being sufficiently low in all samples. Additionally, color stability during a short period of cold storage was studied: it demonstrated that the three tested types of dairy products remained stable throughout the 11-day analysis period with no significant changes. These results prove the potential of the B-phycoerythrin extract as a natural colorant and alternative ingredient to synthetic coloring molecules.

## 1. Introduction

Rhodophyta (or red algae) are microalgae particularly exploited for potential application of their components. *Porphyridium cruentum* (*P. cruentum*) is a red algae from which B-phycoerythrin (B-PE) is obtained. B-PE is a phycobiliprotein in which a phycoerythrobilin chromophore is linked to the protein by a thioether linkage [1]. Phycobiliproteins are water-soluble and colored proteins present in microalgae that play an important role in photosynthesis [2]. Phycobiliproteins are organized in phycobilisomes on the outer surface of thylakoid membranes, and their role is to absorb light in regions where chlorophyll does not, transferring energy to the photochemical reaction centers. In the reaction centers, the electronic energy is then transformed into chemical energy. B-PE is a αβ heterodimer with chains of α-helix structure. Recently, B-PE has gained interest for its application as a fluorescent probe in research and clinical diagnosis and as natural pink colorant in food, fabrics, cosmetics, and drugs [3,4,5]. Only a few studies have used phycoerythrin directly in food application [6,7].

Coloring additives are dyes that confer color to food, drinks, or drugs to which they are added. Color is a driving feature in the choice of food and can deeply condition consumers on what to buy. According to the European Food Safety Authority (EFSA), food colorants are one category of food additives that is used in the food industry to render processed foods more attractive, to restore the color of food after processing, and to give color to otherwise colorless food. Food colorants can be divided in two groups depending on their origin: synthetic and natural colorants [8]. Natural colorants have gained more interest among consumers who demand natural products. In line with an increasing trend of healthy diets and lifestyle, consumers have shifted their preference from synthetic additives towards natural ones. Natural colorants are derived from plants, fruits, or animals and are then extracted and purified. Natural colorants might be sensitive to light exposure, might change color when exposed to different pH, and degrades at high temperatures, and they are generally available in a limited color range. All of these factors limit the use of natural colorants at the industry level. Generally, consumers are more skeptical towards the use of synthetic colorants and prefer natural food dyes for their potential benefits [9]. Therefore, there is a need for new natural colorants for use in food to extend the coloring range of natural food dyes. If possible, food dyes should be extracted from plants or fruits in order to be suitable for all diets.

Several investigations have been made in recent years regarding the use of natural colorants in food and drinks, which include studying aspects of preparation, characterization, application, stability, kinetics, and color evaluation [10,11,12,13,14]. In this context, milk-based products (yogurts, liquid yogurts, and milkshakes) are consumed throughout the world due to their organoleptic and nutritional value. The incorporation of natural colorants to these products can give a better alternative to the use of synthetic molecules by appealing to the growing health-related awareness among consumers [15,16,17,18,19,20]. Other researchers have reported works involving the use of natural colorants in bakery products, soft drinks, and candies [21,22,23,24,25]. Phycobiliproteins have been incorporated in desserts, puddings, sourdough, soft drinks, and dairy products, although their use needs to be further supported by toxicity testing. Moreover, additional research focusing on bioavailability and on the interaction of phycobiliproteins with food, beverages, matrixes, and stabilizers among others is required [26,27,28,29].

In light of the above, we have focused our attention on using a phycoerythrin extract (B-PE extract) as a natural pink colorant, searching for new molecules which can allow for expansion of the natural colorants offered. Another pivotal reason for trying to exploit this dye is its antioxidant effect described in several publications, which adds value to the milk-based products obtained [7,30,31,32]. Phycoerythrin is attractive as an antioxidant and anti-inflammatory drug, having properties to protect against physiological changes under oxidative stress and great effects against aging. Additionally, this compound has fewer side effects than chemical drugs, and algae are eco-friendly sources of this compound and are receiving increasing attention. Hence, the outcome of this study would enhance the organoleptic and nutritional properties of milk-based products using a B-PE extract from red algae.

In the present work, a B-Phycoerythrin extract (B-PE extract) from the red algae *P. cruentum* was studied as a natural colorant in different milk-based products. This study represents a first approach that aims to characterize the B-PE extract by spectroscopy in order to understand the coloring properties of the protein solution. Additionally, a colorimetric study was developed for the color evaluation of samples containing the natural protein extract which reproduce the color of commercially available products.

## 2. Results and Discussion

### 2.1. Characterization of B-Phycoerythrin Extract

The B-PE extract from *P. cruentum* is composed of B-PE, R-phycocyanin (R-PC), and allophycocyanin (APC) [33]. Preliminary analysis for the estimation of phycobiliprotein content in the extract revealed that this colorant extract solution contained 0.8 mg B-PE/mL as the major component (93% *w/w*). Other proteins are present in a significantly minor proportion, such as APC (4% *w/w*) and R-PC (3% *w/w*). The B-phycoerythrin purity ratio (A_545_/A_280_) was 2, which represents a more than sufficient purity grade for its use as a colorant in the food sector.

The absorbance spectrum of the coloring extract shows two main peaks at 545 and 565 nm (Figure 1a), which correspond to B-PE being the main component present in the extract. The absorbance ratio (A_545_/A_280_) was used as a purity indicator. When its value is lower than 4, it indicates the presence of other proteins and that it is not a pure B-PE solution. In this case, R-PC and APC are present in a minor proportion, not having significant repercussions on the final pink color of the B-PE extract.

The protein extract obtained in this study is mostly constituted by B-PE thanks to the removal of contaminant proteins by precipitation with ammonium sulphate. A small shoulder is observed at 620 nm, which corresponds to the R-PC maximum. However, no peak corresponding to APC is observed at 650 nm. Therefore, B-phycoerythrin is the main constituent of the extract, and it confers the characteristic pink color to the extract. Regarding fluorescence, the corresponding spectrum (Figure 1b) shows a maximal fluorescence intensity emission at 575 nm. Both absorption and fluorescence spectra for the B-PE extract demonstrate that B-phycoerythrin is the main protein found and therefore the predominant one in the extract. The other two phycobiliproteins (APC and R-PC) appear in a small proportion.

The pH is a relevant factor that influences the stability of phycobiliproteins. UV-Vis spectroscopy and fluorescence spectroscopy were used to determine the stability of the pink B-PE extract and its optimal pH range of application (Figure 1b). The milk-based products utilized in this work presented pH values of 6.71, 4.21, and 4.16 for milk, liquid yogurt, and yogurt, respectively. Accordingly, it is necessary to know the behavior of the colorant extract inside alimentary samples presenting these pH values.

When the coloring extract is mixed with phosphate buffer at pH values below 3, proteins aggregates form immediately. This is explained by the precipitation of proteins due to the very acidic pH of the solution. A similar behavior was obtained when the B-PE extract was tested in alkaline solutions at pH values above 11: proteins are unstable, and the spectrum of the extract does not retain any shape, being completely disrupted (data not shown). Therefore, the B-PE coloring extract is spectroscopically measured in the range from pH 3.80 to pH 11.0. Every extract sample presents two characteristic picks at 545 and 565 nm, and a third pick is present at 280 nm (proteins total peak). The absorbance spectrum preserves its characteristic profile when the protein extract is in the pH range 4.0–8.0.

B-PE was excited at 465 nm to record the fluorescence spectra of the protein extract at different pH (Figure 1b). At neutral pH, the fluorescence maximum is registered at 575 nm. The spectrum maintains its profile and registers high values of fluorescence intensity (FI) between pH 6.0 and pH 9.0. The intensity of the peak decreases at pH lower than 6 and higher than 9. These results are in accordance with absorbance determinations showing instability in highly acidic or basic media.

Additionally, the B-PE extract was analyzed to obtain color parameters according to the Color Space defined by the International Commission on Illumination (CIELAB-color system: L*, a* and b* values) as a function of pH. The results are summarized in Table 1. Therefore, variations in pH values did not cause important variations in the L*, a*, or b* indexes, showing the stability of the lightness, redness, and yellowness of B-PE extract. This observation is crucial because the solution maintains an attractive pink color in the pH range assayed.

It is important to emphasize that the B-PE extract keeps color parameters unchanged in the pH range assayed and that, when the protein extract is added, coloring parameters are maintained highly similar in the three different groups of milk-based products selected. Therefore, the stability of the redness for the B-PE extract is related to a higher pH stability for B-phycoerythrin biliprotein. Previous works have studied the pH-induced conformational and functional dynamics of B-phycoerythrin isolated from different red microalgae by means of absorption, fluorescence, and other common techniques [33,34]. B-phycoerythrin has a strong functional stability in the pH range 3.5–10, with no remarkable changes in absorbance or fluorescence, suggesting the ability of stable energy transfer or the protein even at low pH values.

In light of the above and taking into account the characterization carried out using absorbance, fluorescence, and colorimetry, applications of this pink B-PE extract may be useful in acidic and neutral food samples, such as in milk-based products.

### 2.2. Milk-Based Products Pigmentation with B-PE Extract

According to the ingredients described on the labels of pink milk-based products found in the market, the most commonly used synthetic dyes are Ponceau 4R (FD&C Red n°1), Bordeaux (FD&C Red nº9), and Allura Red (E-129). Regarding natural molecules, the only one that is usually found in dairy products is carminic acid (E-120) [3,9,16].

Therefore, the color given by the pink B-PE extract to standard products is compared to the color of reference products found in grocery stores which contain other types of colorants, synthetic or natural. It is important to highlight that the current trend in the food industry for human consumption is favoring the use of natural colorants instead of those produced by chemical synthesis.

In coloring studies, the standard product was colored with the pink B-PE extract to achieve a color most similar to the one of the reference product. From coloring assays, a coloring curve is obtained: it plots the difference in color (ΔE*ab) between the sample and the reference against the volume of B-PE extracts added to the standard product (μL). In order to match the color of marketed products in the three types of milk-based products, it was necessary to add a specific volume of B-PE extract to 14 ml of the milk-based product (Table 2). Because the amount of added protein was kept at <50 mg/L in all experiments, its effect on texture and viscosity was negligible, and these results are in agreement with previous studies [15].

In global terms, liquid yogurts required less B-PE extract volume than milkshakes and semisolid yogurts to reproduce the colors of commercial products. The products that required the lowest amount of protein were “H” and “J” liquid yogurts, which obtained staining factors of 3.98 mg/L. On the contrary, the product that required the highest amount of protein was “K” liquid yogurt, which achieved a staining factor of 82.9 mg/L. These results show the potential of the B-PE extract as a natural colorant and its potential economic viability due to the low values of the obtained staining factors.

The progressive addition of the B-PE extract to standard commercial products (white products) promotes the gradual decrease of the ΔE*ab value as the color of the standard progressively approaches the color of the commercial product. The final sample is the one in which the lowest ΔE*ab value is reached (values close to ΔE*ab = 4). It is important to note that, when a ΔE*ab value lower than 5 is reached, the color of the sample is nearly the same as that of the reference when viewed by the naked human eye [15]. Figure 2 shows a vivid presentation of the color samples obtained using the B-PE extract compared to the reference product found in the market. Three representative samples were selected as examples of the milk-based products assayed. As previously commented, the color of each sample was very close to the color of its reference commercial product.

#### 2.2.1. Milkshakes

Six different commercial milkshakes were studied to test the potential coloring capacity of the B-PE extract (Table 2). Thus, as most of the final ΔE*ab values reached were below or very close to 5 (Figure 3), coloring curves obtained from milkshakes suggest that the B-PE coloring extract has good coloring properties.

Milkshake “C” registered the highest staining factor (Table 2), and its coloring curve suggests that a large volume of the B-PE extract (650 μL) was necessary to obtain the final sample (Figure 3). On the other hand, milkshake “F” required the lowest volume of B-PE coloring solution (60 μL) to achieve the desired color. Differently, other milkshakes showed intermediate volumes of the coloring extract to reach colors very close to the ones of commercial products. Staining factors for each sample were calculated as the total amount of B-PE protein needed for each liter (of product) to obtain the same color as that of the reference commercial product (Table 2). Generally, sample staining factors are low, proving that good colors can be obtained with a fairly low quantity of the B-PE coloring extract.

#### 2.2.2. Liquid Yogurts

In the same way, five different commercial liquid yogurts were studied for testing the potential coloring capacity of the B-PE extract. In this case, coloring curves obtained from milkshakes reflect a wide variety of results (Figure 4).

With the exception of product “K”, the rest of the samples required very small extract volumes for achieving the desired color. Thus, staining factors were between 3.98 and 7.15 mg/L from products “J” and “G”, respectively. As an exception, the staining factor of “K” liquid yogurt was quite high (82.9), being 21 times higher than the staining factor of “H” liquid yogurt.

#### 2.2.3. Semisolid Yogurts

Finally, four different pink semisolid yogurts were found in the market and assayed to test the potential coloring capacity of the B-PE extract (Figure 5).

Coloring curves of all assayed semisolid yogurts showed excellent ΔE*ab values. All final ΔE*ab values were equal to or lower than 5, suggesting that the color of each sample was highly similar to the color of its reference commercial product. Overall, staining factors were low, with sample “N” requiring the lowest amount of extract and sample “L” requiring the highest (Table 2).

### 2.3. Stability Studies of Colored Milk-Based Samples

The stability of food products is a crucial aspect to take into account, and because of that, color stability determinations have been developed for milk-based products added with the B-PE extract. Figure 6 shows variations of the color coordinates L*, a*, and b* as well as the color difference ΔE*ab registered for milkshakes, liquid yogurts, and yogurts added with B-PE extract during the storage period of 11 days. The samples selected for each type of milk-based product were obtained by the addition of the B-PE extract to reproduce the color of “D” milkshake, “G” liquid yogurt, and “N” yogurt.

Figure 6c shows the variations in a* index (redness) for the prepared samples. It is possible to appreciate very slight variations, with values remaining practically constant from 0 to 11 days. In this case, the milkshake sample showed the highest values of a* while the yogurt sample registered the smallest.

The liquid yogurt sample had the highest L* and b* values and intermediate values of a* (Figure 6b–d), compared with the milkshake and the yogurt. Moreover, it was observed that values of L* were stable from 0 to 9 days for the three types of milk-based products and slightly increased up to 11 days for liquid yogurts. However, the yogurt sample registered L* values slightly increasing up to 6 days and then remaining constant for 11 days. Despite the above changes, the movements of the values of L* were not observed to have a drastic behavior and ΔL* between 11 days and 0 days did not exceed 1.5 units for all the 3 types of products assayed.

The ΔE*ab values (Figure 6a) were very low for liquid yogurt and milkshake, registering ΔE*ab = 0.25 and ΔE*ab = 0.6 (1 days), respectively. Despite their variations and upward trends, ΔE*ab did not exceed a value of 2.0 for the entire study period. A different trend was observed for yogurt, which started with a value of ΔE*ab = 0.25 at day 1 and concluded with ΔE*ab = 3.0 at 11 days. However, regardless of its high fluctuation, the difference between 11 days and 1 day was 2.75 for yogurt, whereas it was 1.4 and 1.25 for milkshake and liquid yogurt, respectively. It is well known that ΔE*ab values lower than 3.0 cannot be easily detected by the naked human eye and are taken as the same color by consumers [35]. According to these data, it is deduced that the sample color is quite stable and that this fact could be due to the protection afforded by the lipid–protein matrix of milk.

It is well known that the majority of natural pigments have antioxidant activity. Concerning stability studies of colored milk-based samples, the data obtained showed that the sample color is quite stable. This effect could be explained by the antioxidant activity of the B-PE extract as reported by others authors [30,31,32]. The antioxidant effect of phycoerythrin was also shown in a previous study using this colorant in ice-cream [7].

The high stability of the phycoerythrin pigment within the protein–lipid matrix of milk-based products found in this study is in contrast with the statement on storage studies as a generality reported for others authors on storage studies who pointed out that the colors from natural sources tend to lose dyeing strength or to disappear with time [36]. According to them, a greater weight of plant material would be needed for direct use or for extraction of the natural dye compared to the weight of the synthetic dye in order to obtain the same depth of color. Incorporation of natural colorants to food systems faces different challenges such as their relatively low stability in processing and storage conditions and the presence of undesirable odor or flavor characteristics [37].

All this indicates that phycoerythrin maintained its stability throughout the refrigerated storage time for the three types of milk-based products without significant changes in any of the three color coordinates. Values of ∆E*ab ≥ 5.0 were not obtained at any moment during storage. This is the threshold value used by other researchers to indicate the onset of instability [21].

In addition to its antioxidant capacity, another aspect to consider is the toxicological connotation involving phycoerythrin application as a natural colorant. This molecule is gaining more importance in present days as it plays a crucial role in preventing or delaying diseases that involve toxicity in certain organs such as the liver [38]. By and large, artificial antioxidants such as butylhydroxynisole and butylhydroxytoluene possess potential health risk and toxicity, which makes it crucial to substitute them with new harmless natural antioxidants like phycoerythrin.

## 3. Materials and Methods 

### 3.1. Production of Phycoerythrin Extract

The B-phycoerythrin extract was obtained using the corresponding methodology [34,39] from cells of the microalgae *P. cruentum*, generously provided by the Chemical Engineering Department of Almería University (Spain). The phycobiliprotein solution was extracted from the lyophilized microalgae using phosphate buffer 0.1 M pH 5.5 for reconstitution. The intracellular material was obtained by disrupting the cell wall by osmotic shock. The sample was centrifuged (Medtronic BL-S, P-Selecta, Barcelona, Spain) to recover the phycobiliproteins in the supernatant. The supernatant was precipitated with 60% saturation ammonium sulphate ((NH_4_)_2_SO_4_). The precipitate was centrifuged, and the pellet was recovered and resuspended in a small volume of the suitable buffer. The protein solution was dialyzed to obtain the final B-phycoerythrin extract. The concentrated protein solution was placed in a clean container, and 1% *w/w* of sodium azide (NaN_3_) was added as a preservative. The sample was stored at 4 °C.

### 3.2. Characterization of the B-Phycoerythrin Extract

Absorbance spectra were recorded on the UV-visible spectrophotometer (Lambda 20, Perkin Elmer, software UV Data Manager, Waltham, Massachussetts, U.S.) between 250 and 750 nm. Fluorescence spectra were recorded on the spectrofluorimeter (FP-6500, Jasco, software Spectra Manger, Madrid, Spain) between 500 and 700 nm. Recording the absorption and fluorescence spectra of aqueous solutions in a large pH range served to monitor the influence of pH on the stability of the spectral parameters of the B-phycoerythrin extract. The protein extract was dissolved in 20 mM phosphate buffer at different pHs (ranging between 1.0 and 13.0) to achieve an appropriate concentration. Absorption spectra of the B-PE extract were determined after gradual falls or increases in pH (pH 7–1 and pH 7–12), and the stability was monitored by measuring the absorbance at 545 nm. Fluorescence spectra were determined with the same solutions, and a fluorescence intensity at 576 nm was registered. All spectra were recorded at room temperature.

The amounts of different phycobiliproteins (B-PE, R-PC, and APC) in the B-PE extract were calculated from measurements of the absorbance at 280, 545, 565, 620, and 650 nm using the following equations [33]:R-PC (mg/mL) = (A_620 nm_ − 0.7·A_650 nm_)/7.38(1)
APC (mg/mL) = (A_650 nm_ − 0.19·A_620 nm_)/5.65(2)
B-PE (mg/mL) = (A_565 nm_−2.8·[R-PC] − 1.34·[APC])/12.7(3)

### 3.3. Alimentary Samples

Alimentary samples are constituted by milk-based products (milkshakes, liquid yogurts, and semisolid yogurts) available in grocery stores. The selected commercial products have a pink color; while standard products do not carry any coloring additive and therefore have a white color. Regarding milkshakes, semi-skimmed white milk of the same brand was used as a standard. On the other hand, for liquid yogurts and semisolid yogurts, a natural white liquid and semisolid yogurt of the same brand were used as a standard. Different commercial products were utilized to simulate colors using the B-phycoerythrin extract: six milkshakes, five liquid yogurts, and four yogurts. Storage of the samples was done in a domestic refrigerator (Fagor, 3Ffk-6625X, Mondragón, Spain) at T = 5 ± 2 °C.

Only semisolid yogurts were manipulated before colorimetric analysis due to their consistency. They were weighed and then placed on the magnetic stirrer (HB502, Bibby, Stone, UK) for 10 min to obtain a liquid matrix. The pink protein extract was added to the matrix under constant stirring to allow its homogenous mixing.

### 3.4. Color Analysis

The color of the samples was analyzed by a colorimeter (CM-5, Konica Minolta, data software CM-S100W SpectraMagic TM NX, Version 2.2, Osaka, Japan) operated in the transmittance mode. The color was registered according to the CIELAB color space, and the color space values were obtained: L* represents the light-dark pair (lightness), a* represents the red-green pair (redness) and b* represents the yellow–blue pair (yellowness). The recommended illuminant/observer pair D65/10° was used following the method of measurement for translucent semisolid samples [15].

First, the color of the reference product (commercial product) was registered by the colorimeter, followed by registration of the color of the standard sample (sample to be added with B-PE extract). Successively, few microliters of B-PE extract were added to the standard sample, and the color change was registered. After each B-PE extract addition, the color difference between the alimentary commercial product and the colored sample was defined by ΔE*ab. For determination of the CIELAB color difference ΔE*ab, the following equation was used:∆E*ab = [(∆L^*^)^2^ + (∆a^*^)^2^ + (∆b^*^)^2^]^1/2^ = [(L^*^_i_ − L^*^_0_)^2^ + (a^*^_i_ − a^*^_0_)^2^ + (b^*^_i_ − b^*^_0_)^2^]^1/2^(4)
where L^*^_0_, a^*^_0_, and b^*^_0_ were the values of the alimentary commercial product and L^*^_i_, a^*^_I_, and b^*^_i_ were the measured values of each sample added with the B-PE extract.

## 4. Conclusions

The present study demonstrates that the B-PE extract can be used as a coloring additive for milk-based products. The color of commercial reference products containing synthetic and natural colorants was compared to the color of samples dyed with the pink B-PE extract. Colors and color differences were measured according to the CIELAB color space, and the colorimetry-based approach was suitable for the purpose of the study. Coloring assays conducted using a colorimeter prove that the colors of the samples are similar to the colors of the reference samples taken from the market. Staining factors determine that low amounts of B-PE are necessary to obtain acceptable samples colors. pH influences the spectroscopic properties of the protein extract and the stability of B-PE. The optimal pH range for stability of the B-PE extract isolated from *P. cruentum* is 4.0–9.0. The study proves that the B-PE coloring extract could be an interesting natural colorant for its attractive coloring properties, and it could expand the list of natural coloring additives now available on the market.

## Figures and Tables

**Figure 1 molecules-26-00297-f001:**
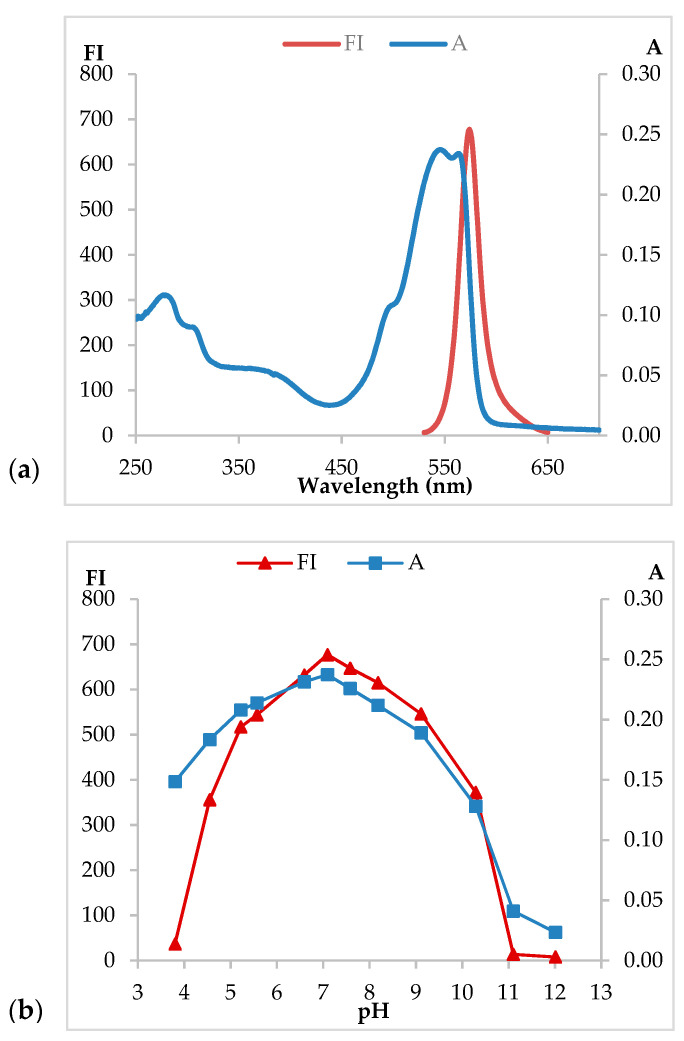
(**a**) Absorption and fluorescence spectra of the B-phycoerythrin (B-PE) extract from *P. cruentum* at pH = 7.0, and (**b**) variation of absorbance (■) (at 545 nm) and relative fluorescence intensity (▲) (at 575 nm) of the B-PE extract solution at various pH values in 20 mM phosphate buffer: for absorbance and fluorescence measurements, the protein concentration was 19·10^−3^ mg B-PE/mL.

**Figure 2 molecules-26-00297-f002:**
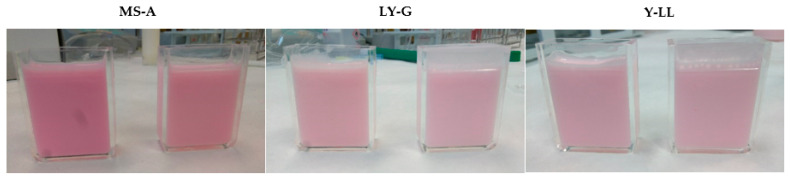
Photographs showing a vivid presentation of the market products and milk-based products obtained using the B-PE extract as a natural colorant: in this figure, three representative samples are shown: sample A, representing milkshakes (MS-A); sample G, representing liquid yogurts (LY-G); and sample LL, representing yogurts (Y-LL). In each pair of photos, on the left is the reference product found in the market and on the right is the sample obtained using the B-PE extract.

**Figure 3 molecules-26-00297-f003:**
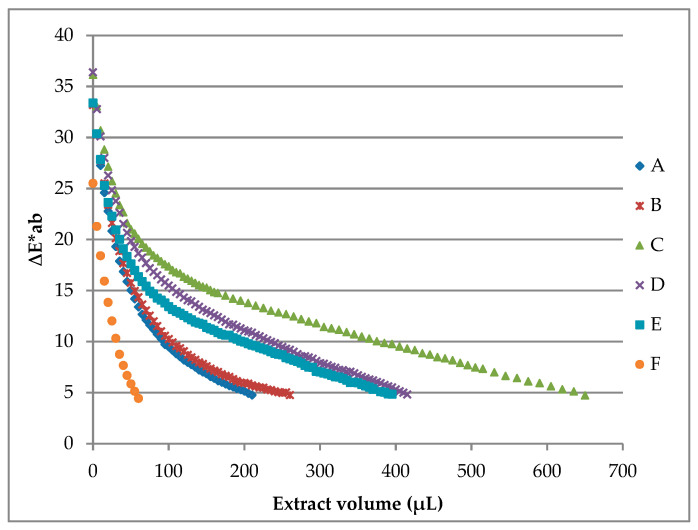
Coloring curves of milkshakes resulted from coloring assays using the B-PE extract from *P. cruentum* ([B-PE] _extract_ = 1.12 mg B-PE/mL).

**Figure 4 molecules-26-00297-f004:**
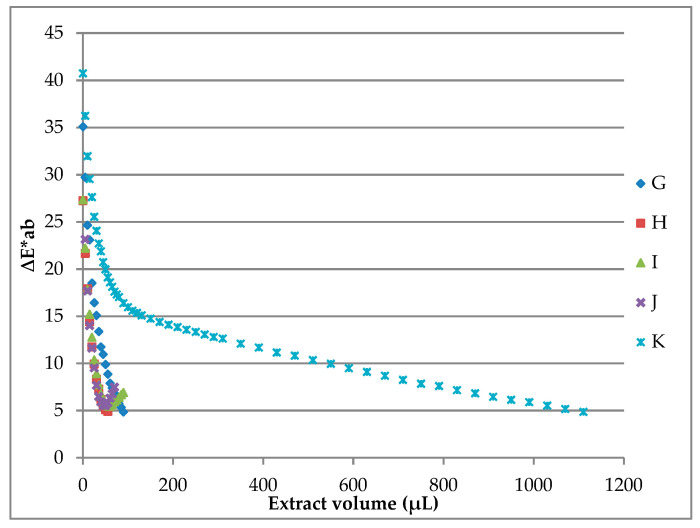
Coloring curves of liquid yogurts resulted from coloring assays using the B-PE extract from *P. cruentum* ([B-PE] _extract_ = 1.12 mg B-PE/mL).

**Figure 5 molecules-26-00297-f005:**
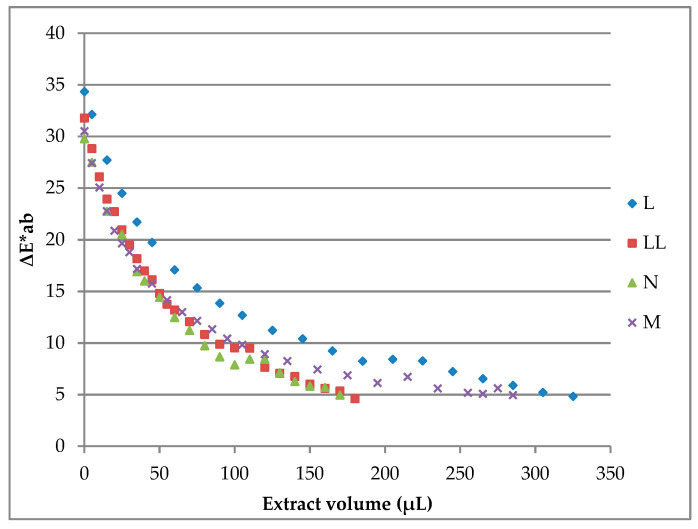
Coloring curves of liquid yogurts resulted from coloring assays using the B-PE extract from *P. cruentum* ([B-PE] _extract_ = 0.8 mg B-PE/mL).

**Figure 6 molecules-26-00297-f006:**
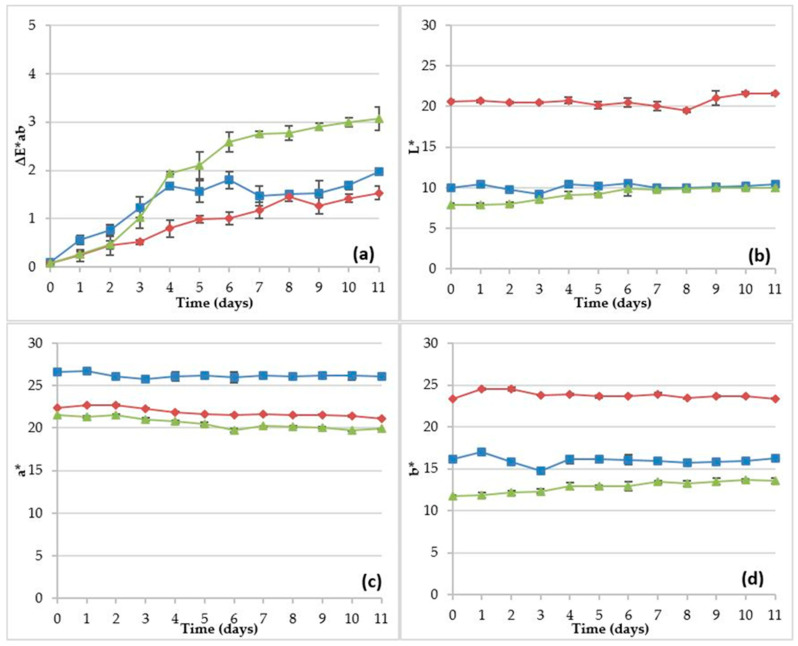
Changes in color indexes (L*, a*, b*, and ∆E*ab are the CIELAB color system values) of milk shake (■), liquid yogurt (♦), and yogurt (▲) pigmented with B-PE extract during a short period cold storage: the values plotted are the means of triplicate ± standard deviations.Changes in ∆E*ab (**a**); changes in L* (**b**); changes in a* (**c**) and changes in b* (**d**).

**Table 1 molecules-26-00297-t001:** Color parameters (CIELAB-color system) for the B-PE extract at different pH values (media ± SD). L*, a* and b* are the three CIELAB values to defining the color.

PH	L*(D65)	a*(D65)	b*(D65)
3	97.85 ± 0.02	4.28 ± 0.02	−2.82 ± 0.02
4	97.85 ± 0.02	4.37 ± 0.03	−2.75 ± 0.02
5	97.80 ± 0.02	4.54 ± 0.05	−2.83 ± 0.03
6	97.80 ± 0.03	4.49 ± 0.05	−2.77 ± 0.03
7	97.81 ± 0.02	4.58 ± 0.06	−2.77 ± 0.04
8	97.88 ± 0.15	4.39 ± 0.30	−2.55 ± 0.15
9	97.94 ± 0.04	4.33 ± 0.09	−2.48 ± 0.05

**Table 2 molecules-26-00297-t002:** Staining factors for different milk-based products: these data are obtained according to the coloring curves obtained for each product (Figure 2, Figure 3 and Figure 4).

Product	Commercial Brand	Extract Volume Added (μL)	∆E*ab Reached	Staining Factor (mg/L)
Strawberry Milkshakes Extract Concentration 1.12 mg B-PE/mL	A	205	4.93	16.12
	B	260	4.77	20.36
	C	650	4.73	49.55
	D	410	5.00	31.78
	E	390	4.87	30.27
	F	60	4.43	4.77
Strawberry Liquid Yogurts Extract Concentration 1.12 mg B-PE/mL	G	90	4.85	7.15
	H	50	4.89	3.98
	I	65	5.45	5.17
	J	50	5.53	3.98
	K	1120	4.87	82.90
Strawberry yogurts Extract concentration 0.8 mg B-PE/mL	L	325	4.82	17.97
	LL	180	4.59	10.05
	M	285	4.95	15.80
	N	170	4.95	9.50

## Data Availability

Not applicable.

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
