# Peer review of "Using a B-Phycoerythrin Extract as a Natural Colorant: Application in Milk-Based Products"

_molecules, 2021, doi:10.3390/molecules26020297_

Round 1
Reviewer 1 Report
Edible dyes from natural products are useful for the food industry. The work here is presenting new edible natural-product dyes. But there are several questions before considering its publishing.
1, there are already kinds of food-derived dyes, the authors should provide better comparison and experimental proof or so to show the necessity and essential reason to exploit these dyes in this work.
2, Photos should be shown to give a vivid presentation of the dyes and their blends with milk drinks.
3, Microscopies should be done to show the micro-level morphologies of the blends with milk, as well as the characterization of their long-term stability.

Reviewer 2 Report
The article has a good structure. It's really easy to read. We can follow easily the methodology to characterize and qualify the new natural dyes. Methods, results, curves are well presented. We have a well defined interpretation for each result in the article. The curves are very clear and well defined. Concerning figure 5, it is better to identify each graph in the text.
Reviewer 3 Report
The manuscript entitled "Using a B-Phycoerythrin extract as a natural dye: application in milk products" is a good job, well written.
In literature there are many works concerning B-phycoerythrin in food as a natural coloront.
The topic is topical, finding new pigments of natural origin is very important. Personally the authors could think of adding more scientific information to give more weight to the manuscript: toxicological aspects, and especially the antioxidant activity of the pigment, since most pigments have this activity, and certainly in the results obtained the non-variation of the color is due to this, but it is not reported.
I suggest the authors to write it in the text with the bibliographic references.
Round 2
Reviewer 3 Report
the authors made significant revisions.